# Clinical Evaluation Tool for Vascular Health–Endothelial Function and Cardiovascular Disease Management

**DOI:** 10.3390/cells11213363

**Published:** 2022-10-25

**Authors:** Fang Wen, Yue Liu, Hongyu Wang

**Affiliations:** 1Department of Vascular Medicine, Peking University Shougang Hospital, Beijing 100144, China; 2Vascular Health Research Center of Peking University Health Science Center (VHRC-PKUHSC), Beijing 100191, China; 3Key Laboratory of Molecular Cardiovascular Sciences (Peking University), Ministry of Education, Beijing 100191, China; 4Beijing Shijingshan District Key Clinical Specialty of Vascular Medicine, Beijing 100144, China; 5Heart and Vascular Health Research Center of Peking University Clinical Research Institute (HVHRC-PUCRI), Beijing 100191, China; 6Intellectualized and Digital Heart and Vascular Health Management Instiute of Integrated Traditional Chinese Medicine and Western Medicine, Chengdu Xindu District People’s Hospital, Chengdu 610500, China

**Keywords:** cardiovascular disease, Beijing vascular health stratification, vascular health, endothelial function

## Abstract

There are 330 million people suffering from cardiovascular diseases (CVD) in China, and two out of every five deaths were due to CVD. CVD has become the main disease burden in China. Vascular health management can detect subclinical vascular diseases such as endothelial dysfunction. Through controlling risk factors, vascular function, such as endothelial function, can be improved and cardiovascular events can be prevented from the upstream. Peking University Shougang hospital is the first practitioner of life-long vascular health management since 2010 in China. The established Beijing Vascular Health Stratification (BVHS) focuses on the comprehensive evaluation of vascular health function and structure and explores the application of information technology and artificial intelligence in vascular health management. The life-long vascular health management and tertiary hospital–primary hospital–family service model guided by BVHS can better realize the prophylaxis of CVD. The prevention and control strategy of CVD based on information technology and vascular health, especially endothelial function management, can help to implement the “healthy China 2030” plan. In this review, we focus on advances in the clinical assessment of vascular endothelial function, including the evaluation of endothelial function, the evaluation of arteriosclerosis, new potential biological markers to provide new possible therapeutic targets, and BVHS, a comprehensive vascular aging assessment system. Strengthening the assessment of cardiovascular health and endothelial function is of great significance for the occurrence of cardiovascular diseases in risk groups and the occurrence of adverse events in patients with cardiovascular diseases.

## 1. Introduction

Systemic and pulmonary circulations constitute a complex organ that serves multiple important biological functions. Consequently, any pathological processing affecting the vasculature can have profound systemic ramifications. Endothelial and smooth muscle are the two principal cell types composing blood vessels. Critically, endothelial proliferation and migration are central to the formation and expansion of the vasculature, both during embryonic development and in adult tissues. Endothelial populations are quite heterogeneous and are both vasculature type- and organ-specific. There are profound molecular, functional, and phenotypic differences between the arterial, venular, and capillary endothelial cells and the endothelial cells in different organs. Given this endothelial cell population diversity, it has been challenging to determine the origin of endothelial cells responsible for the angiogenic expansion of the vasculature. Recent technical advances, such as precise cell fate mapping, time-lapse imaging, genome editing, and single-cell RNA sequencing, have shed new light on the role of venous endothelial cells in angiogenesis under both normal and pathological conditions. Emerging data indicate that venous endothelial cells are unique in their ability to serve as the primary source of endothelial cellular mass during both developmental and pathological angiogenesis and CVD management [1].

Prevalent cases of total CVD nearly doubled from 271 million in 1990 to 523 million in 2019, and the number of CVD deaths steadily increased from 12.1 million in 1990, reaching 18.6 million in 2019. The global trends for disability-adjusted life years (DALYs) and years of life lost also increased significantly, and years lived with disability doubled from 17.7 million to 34.4 million over that period [2]. The China Cardiovascular Health and Disease Report 2021 pointed out that there are 330 million cardiovascular disease patients in China, and the prevalence of cardiovascular disease continues to rise, which has become the primary cause of increased morbidity and mortality [3]. The main pathophysiological mechanism of heart, brain, kidney, and other tissue and organ diseases is the structural and functional impairment of the supply vessels and the accelerated aging process of the vessels, resulting in adverse vascular events such as CHD, stroke and dementia, peripheral occlusives disease, and even sudden death [4]. Therefore, the intervention and dynamic monitoring of alternative vascular health indicators are important strategies for the prevention of cardiovascular and cerebrovascular diseases. These indicators can effectively reflect the clinical endpoint of the progression of vascular lesions caused by risk factors, thereby maintaining vascular health and slowing vascular damage.

In 2004, we put forward the concept of the vascular disease early detection technology system as the basis of systemic vascular disease prevention and treatment, which was approved by the former Ministry of Health of the People’s Republic of China’s ten-year 100 projects and promoted to the whole country. The program was the first government-driven nationwide vascular disease prevention program in the world, and its experience was disseminated through various means, such as the Annual meeting of the Chinese International Congress of Vascular Medicine (CCVM) and the concurrent launch of the Chinese Heart and Vascular Health Promotion Program (CHVHPP). In 2010, Peking University Shougang Hospital established China’s first vascular medicine Center with the goal of “life-long management of vascular health”. In 2019, the Peking University Health Science Center approved the establishment of a Vascular Health Research Center featuring innovation and interdisciplinary research. Our research on vascular function and clinical practice of comprehensive diagnosis and treatment of vascular diseases from 1997 to 2022, especially the series of studies on vascular function, have confirmed that health management centering on the whole life-cycle of blood vessels and early detection and treatment of vascular aging-related diseases will improve the prevention and treatment of CVD [5,6,7,8,9].

## 2. Whole-Life-Cycle Vascular Health Management—“Shougang Vascular Health Management Model”

It is of great practical significance to take vascular health management as the starting point to prevent vascular damage. The hierarchical management of different vascular risk groups can also save medical resources. The Vascular Medical Center of Peking University Shougang Hospital has an integrated model of tertiary hospital and community medical service centers which facilitates the whole-course management of CVD. We are actively exploring the insertion of the “AI-based intelligent diagnosis and treatment system for cardiovascular and cerebrovascular diseases”. The system can provide clinical doctors with intelligent collaborative auxiliary diagnosis and treatment of cardiovascular and cerebrovascular diseases. The application of this technology to community health service centers can greatly improve the clinical diagnosis and treatment skills of general practitioners. In 2010, the Peking University Shougang Hospital Vascular Medicine Center was established. It combines vascular health community management, early detection and reversal of vascular diseases, vascular disease intervention and surgical treatment, and explores the lifelong vascular health management model integrating prevention, diagnosis, treatment, and rehabilitation in tertiary hospitals and communities— the “Shougang Vascular Health Management Model”. The center also implements strategies for the early prevention and rehabilitation of vascular diseases that integrate traditional Chinese and Western medicine. Vascular medicine has become a comprehensive medical specialty covering general practice, preventive medicine, clinical medicine, rehabilitation medicine, and medical humanities [5]. The Beijing Vascular Health Stratification (BVHS) provides a feasible standard for whole life-cycle management of vascular health and damage [10]. Finally, we established the concept of “first diagnosis to community, serious disease to hospital, Guided by the policy and concept of “rehabilitation to community”, and the hospital provides patients with two-way referral, remote consultation, posthospital follow-up, etc., which really realizes the mutual transfer between the upper and lower hospitals in the medical alliance [11].

## 3. Vascular Health and Endothelial Function Detection for the Population

Vascular function tests and vascular imaging tests are useful for assessing the severity of atherosclerosis. Since vascular dysfunction and vascular morphological alterations are closely associated with the maintenance and progression of atherosclerosis, vascular tests may provide additional information for cardiovascular risk assessment. Measurement of the ankle brachial index (ABI) has been performed not only for screening for peripheral artery disease but also for cardiovascular risk assessment in clinical practice. However, the ABI method does not always provide reliable data because the ABI value is falsely elevated despite the presence of occlusive arterial lesions in the lower extremities of patients with noncompressible lower limb arteries, which can lead to incorrect cardiovascular risk assessment. Pulse wave velocity (PWV), an index of arterial stiffness for risk assessment, showed that higher PWV was significantly associated with a higher risk of cardiovascular events, cardiovascular mortality, and all-cause mortality in patients with a history of coronary artery disease (CAD) or stroke. Vascular tests are also useful to achieve a better understanding of the underlying pathophysiology of cardiac disorders. Flow-mediated vasodilation (FMD), an index of endothelial function, endothelial function assessed by the reactive hyperemia index (RHI), and impaired arterial stiffness assessed by carotid–femoral PWV (cfPWV) and carotid intima–media thickness (IMT) was significantly related to cardiovascular health. These findings indicate the importance of lifestyle modifications for maintaining vascular function and preventing the development of hypertension and the progression of atherosclerosis [12]. The applicable population for endothelial function assessment should include: (i) At least 14 years old; (ii) Family history of early onset of cardiovascular and cerebrovascular diseases: family history of cardiovascular and cerebrovascular diseases or atherosclerosis, especially early onset in direct relatives (male: 55 years old, female: 65 years old); (iii) Have symptoms such as long-term dizziness, chest tightness after activity or in resting state, palpitations, and intermittent claudication, which have not been clearly diagnosed; (iv) Patients who have been diagnosed with hypertension (including critical hypertension), hyperlipidemia, hyperuricemia, hyperhomocysteine, diabetes (including elevated fasting blood glucose and impaired glucose tolerance), or who have high risk factors of cardiovascular and cerebrovascular diseases such as obesity, long-term smoking, high-fat diet, insomnia, and lack of physical exercise; (v) Patients with clear history of CHD, stroke, ischemic kidney disease, LEAD, ischemic bowel disease, etc., were evaluated for therapeutic effect and prevention of recurrence of vascular events.

## 4. Evaluation Parameters of Endothelial Function

### 4.1. Vascular Endothelial Function Assessment

Vascular endothelial cells are highly active monolayer cells, and by adjusting the vasodilatation and contraction, growth inhibition and growth-promoting anti-inflammatory and proinflammatory functions, the balance between antioxidants, and the promotion of oxidation to maintain tension and the structure, the blood vessels respond to fluids and the stimulation of the nervous system, especially the hemodynamic. They can synthesize and release vasoactive substances to regulate vascular tone, regulate platelet function, inflammatory response, and vascular smooth muscle cell growth and migration, and play important roles in the pathological progression of vascular diseases. Endothelial dysfunction is associated with atherosclerosis and CVD and is associated with multiple cardiovascular risk factors. Early detection and reversal of vascular endothelial dysfunction and the maintenance of vascular endothelial health can help prevent heart and vascular diseases. In the early stage of arteriosclerosis, there will be symptoms of vascular endothelial dysfunction. Exercise, weight loss, drug therapy, and other methods can effectively improve this situation. Low nitric oxide (NO) production is a cause and consequence of endothelial dysfunction [13,14]. Species of reactive oxygen species (ROS) are involved in maintaining vascular endothelial homeostasis. Inflammation, oxidative stress, and autonomic dysfunction can reduce vascular endothelial function [15]. The vascular endothelium can release some biomarkers when stimulated. Both serum biomarkers indicating the impaired endothelium and endothelial function detection indexes can assess the function of vascular endothelium.

#### 4.1.1. Coronary Artery Endothelial Function Assessment

Coronary endothelial dysfunction is generally assessed by an invasive method which is directly measured by coronary angiography and Doppler guide wire. In patients with CHD, injection of acetylcholine through the coronary artery induces abnormal coronary artery vasoconstriction. This technique is suitable for patients requiring coronary angiography. However, invasive coronary angiography to assess vascular function in asymptomatic patients is often difficult to accept. Recent years have seen the emergence of noninvasive functional tests to assess coronary microvascular function, such as positron emission tomography (PET) [16], myocardial perfusion imaging, blood-oxygen-level-dependent magnetic resonance imaging, echocardiography, intravascular ultrasound (IVUS), and optical coherence tomography (OCT).

A PET scan of the heart utilizes a radiotracer (82 rubidium or 13 N-ammonia for rest and stress perfusion) [17]. A substance such as glucose, protein, nucleic acid, fatty acid, etc., labeled with a short-lived radionuclide (such as 18F, 11C, etc.) is injected into the human body, and the metabolic activity of life is reflected by the aggregation of the substance in metabolism so as to achieve the purpose of diagnosis. 18F-Fluorodeoxyglucose (FDG) is independently associated with the severity of CAD and with the occurrence of AMI [18]. In a high cardiovascular risk population, NaF atherosclerotic plaque uptake was related to the burden of cardiovascular risk factors and thoracic fat volume [19]. Oxygen-15-labeled water is a freely diffusible and metabolically inert tracer. Myocardial perfusion reserve (MPR) is clinically assessed by quantitative [O-15]H_2_O PET to reflect the ability of the vascular bed to increase perfusion and microvascular responsiveness [20]. Moreover, 13 N-ammonia PET imaging, which measures MBF, is used to assess improvements in microvascular function [21].

In a study showing a link between post-traumatic stress disorder and ischemic heart disease, we assessed myocardial blood flow reserve (MFR) by PET, which can indicate coronary microvascular function in the absence of significant coronary artery stenosis [22]. Coronary MRI can measure subcardial and epicardial perfusion to assess potential coronary microvascular dysfunction in patients with nonobstructive coronary artery disease [23]. Quantitative parameters of echocardiography in chronic heart failure patients are related to vascular endothelial function, and their combination can effectively predict the risk of MACE in the near future, providing reference for clinical treatment [24]. Intravascular ultrasound (IVUS) is a diagnostic method that combines noninvasive ultrasound technology and invasive catheter technology, providing cross-sectional images of 360° views of blood vessels which can accurately grasp the morphology and degree of stenosis of the blood vessels. Coronary optical coherence tomography (OCT) is a diagnostic imaging study within the coronary arteries that helps to show the microscopic structure of normal and diseased arteries, identifying the formation of calcified plaques and neonatal intima after stent placement [25].

#### 4.1.2. Flow-Mediated Dilation of Brachial Artery (FMD)

FMD is a noninvasive method for detecting early abnormalities in vascular endothelial function. Vascular ultrasound equipment can be used to detect the brachial artery diameter after baseline and cuff band occlusion of blood flow and release. The calculated change rate of brachial artery diameter is FMD, FMD = (internal diameter after arterial reactive congestion–internal diameter of arterial base)/internal diameter of arterial base × 100%, the normal value of FMD > 10%. The test is simple, noninvasive and reproducible for the early detection of vascular disease and is widely used in clinical studies as a surrogate endpoint and prognostic indicator for assessing the risk of CVD. FMD is highly dependent on the technical level of the tester and is susceptible to interference from a variety of factors, and ultrasound imaging is not suitable for high-precision detection. Therefore, the accepted operational criteria for clinical use of FMD tests need to be further defined. The endothelial cells respond to blood flow-induced increases in vascular wall shear stress by increasing vasodilatory autocoid synthesis, leading to vascular smooth muscle relaxation. In present studies, researchers detected the response to transient and sustained shear stress [26]. The limitation of FMD is that it is interfered with by immediate factors such as examination time, the location of blocking blood flow lacks uniform standardization, and the measurement results are affected by the skill level of the operator.

#### 4.1.3. Peripheral Arterial Tonometry (PAT) Detection

PAT detection of the fingertip is similar to FMD in that optical plethysmography is used to measure the amplitude of the finger pulse wave, which reflects the microcirculation function of small blood vessels. PAT is a noninvasive method for assessing microvascular endothelial function, reflecting changes in finger pulse volume amplitude during reactive hyperemia, and assessing endothelial function by analyzing finger pulse wave amplitude. Reactive hyperemia–peripheral arterial tonometry (RH-PAT) can be used to calculate the endothelial function index and the reactive arterial congestion index (RHI). The RH-PAT index is highly expected in the evaluation of the efficacy of the first preventive screening and cardiac secondary prevention interventions because of its simplicity [27].

PAT is associated with multiple traditional cardiovascular risk factors, such as male sex, body mass index, waist circumference, LDL cholesterol, diabetes, smoking, hypertension, and a family history of CHD. It can also be used as a noninvasive peripheral blood intraductal function measurement method to study the predictive value of cardiovascular events. Compared with other examinations, PAT has many advantages, such as being noninvasive, a simple operation, having repeatable results, etc., and can be used as a tool for screening and prognostic assessment. However, PAT also has defects, such as the influence of peripheral vascular bed blood flow by autonomic tension and environment, measurement pulse changes in volume rather than blood flow, probe sensitivity to movement may lead to errors, etc. In addition, the fingertip detection device uses a disposable finger sleeve which is slightly higher in clinical and research costs.

### 4.2. Pulse Pressure (PP) Reflecting Vascular Health and Endothelial Function

PP is the difference between systolic pressure and diastolic pressure, it is used to check a kind of method of blood pressure. With the aging process, the elasticity of arterial wall decreases and stiffness increases and the phenomenon of high systolic blood pressure, diastolic decreasing, and pulse pressure increasing is easy to occur. Increased arterial pressure and reduced arterial wall elasticity associated with vascular aging can lead to cardiovascular disease. Pulse pressure = systolic blood pressure–diastolic blood pressure (mmHg), 3.3. Formatting of Mathematical Components, the reference range is 30~40 mmHg [28]. PP is also a raw index of arterial stiffness (Table 1).

### 4.3. Ultrasonography Was Used to Assess Arterial Structural and Functional Abnormalities Reflecting Vascular Senescence

Systemic arterial lesions and vascular senescence can be diagnosed and evaluated by ultrasound to assess whether endo-media thickening, plaque formation, and lumen stenosis or occlusion [36]. Carotid ultrasound can not only clearly display the thickness, plaque formation, location and size, degree of vascular stenosis and occlusion, but also accurately measure and locate, and analyze, the hemodynamic results of the detected arteries. Criteria for determination of atherosclerotic plaque: echo structures with protrusion of more than 0.5 mm, abnormal blood flow defects with protrusion of more than 0.5 mm, or local intima–media thickness of carotid artery more than 1.5 mm or more than 50% of the thickness of adjacent intima–media were observed in both longitudinal and cross-sectional scanning of vessels.

Assessment of extracranial carotid plaque (CP), carotid artery stenosis or occlusion, and carotid intima–media thickness (CIMT) can be used as an observation window of atherosclerosis in the vascular bed of the whole body, and it can dynamically and continuously observe the progress of early atherosclerotic lesions, which is easy to operate, cheap, and reproducible. CIMT and CP, subclinical atherosclerosis indicators obtained by carotid ultrasound examination, can be used as early assessment indicators of vascular disease and can predict future CVD risk, and the predictive value of CHD risk can be improved by adding CIMT and CP to traditional risk factors [37]. Conventional ultrasound of subclavian artery, abdominal aorta, renal artery, and lower limb artery can observe the inner diameter of the artery, whether the intima is smooth, measuring the thickness of the intima–media, whether there is plaque formation and the shape, size, and echo strength of the plaque, and whether the lumen is narrow or occlusive. Color Doppler flow imaging was used to observe the intraluminal blood flow filling, pulse Doppler spectrum, and peak velocity.

Atherosclerotic plaques can become unstable, rupture, or erode, which leads to cardiovascular events. Plaque stability is related to the level of inflammatory cells and the thickness of the cap. Atherosclerosis is not a classic autoimmune disease. Elevated glucose level, dyslipidemia, and other metabolic alterations that accompany the disease development are tightly involved in the pathogenesis of atherosclerosis at almost every step of the atherogenic process via inflammation [38]. The majority of CD4+ T cells in atherosclerotic lesions respond to oxLDL to produce inflammatory cytokines, which exhibited atherosclerosis also is a chronic inflammatory disease with a secondary autoimmune component [39]. The limitation is that it depends on the skill level of the operator and cannot measure the thickness of the outer membrane of the blood vessel [40].

### 4.4. Evaluation of Arteriosclerosis Reflecting Endothelial Function

Arterial stiffness is excessive fibrosis and decreased elasticity in the process of vascular aging, accompanied by collagen deposition, rupture or the degradation of elastic fibers, arterial middle layer necrosis, calcification, and collagen crosslinking. Activation of the renin–angiotensin system, the initiation of oxidative stress, increased salt intake, and genetic factors can cause excessive arterial fibrosis and extracellular matrix deposition, thus accelerating aging-related vascular injury and sclerosis. The process of accelerated arterial aging is not only the result of the joint action of risk factors such as hypertension, hyperlipidemia, diabetes, and hyperuricemia, but also the root cause of hypertension and diabetes. Arterial stiffness mainly depends on the functional state of the great arteries. Aging and blood pressure are the main determinants, resulting in decreased synthesis and increased degradation of elastin, while increased synthesis and degradation of type III and type I collagen.

#### 4.4.1. Pulse Wave Velocity (PWV) Evaluation

PWV is a common assessment of arterial stiffness and can be calculated by measuring pulse wave conduction distance and time between two arterial segments. PWV can be measured by vascular ultrasound equipment or automated vascular detection devices and can be detected in different arterial segments, such as carotid–femoral, carotid–radial, or brachial–ankle. Carotid–femoral artery pulse wave conduction velocity measurement is currently considered as the gold standard for the evaluation of arterial stiffness. Arteriosclerosis is not only a manifestation of vascular aging, but also a predictor of CVD risk. Higher arterial stiffness is significantly associated with an increased burden of subclinical disease in seven coronary arteries, lower limb arteries, and cerebrovascular vessels. In addition, the process of vascular sclerosis associated with aging is accelerated by a variety of risk factors. In addition to the adverse consequences of traditional cardiovascular events, vascular cognitive impairment is another important hazard. Unlike Alzheimer’s disease, its patients have shorter life expectancy and heavier social and family burden. PWV can be used as an important marker of vascular injury to assess the overall risk of vascular aging and injury in patients and facilitate early identification of vascular injury [41,42].

Arterial stiffness reflects the degree of arteriosclerosis and it is considered to be a good predictor of cardiovascular events. It also is an index of vascular function in conjunction with cardiac function. Arterial stiffness is physically represented by an elastic modulus, but it is not easy to measure in blood vessels in vivo. It has been shown that the elastic modulus of blood vessels is related to PWV, and PWV has become widely used around the world as a surrogate marker of arterial stiffness since it is relatively easy to measure. The drawback of PWV is its essential dependence on blood pressure at the measuring time [43]. The limitation of PWV is that it is heavily affected by blood pressure. The accuracy of recording pulse waveforms in patients with atrial fibrillation or peripheral artery disease is unreliable [44].

#### 4.4.2. Detection of Cardio Ankle Vascular Index (CAVI)

CAVI is a new evaluation index of arterial stiffness derived from the stiffness coefficient β and which is also related to arterial compliance. A series of studies have shown that CAVI can reflect vascular health and aging, and is associated with arterial damage in patients with hypertension, diabetes, metabolic syndrome, and dyslipidemia, thus becoming a useful indicator for evaluating vascular aging [45]. The CAVI was developed based upon the β formula. CAVI reflects the arterial stiffness of the arterial tree from the origin of the aorta to the ankle arteries. CAVI = $a\times \left(2\rho \times \frac{ln\frac{Ps}{Pd}}{\Delta P}\times PWV^{2}\right)+b$. *Ps* = systolic blood pressure; *Pd* = diastolic blood pressure; ρ = blood density, equation. Δ*P* = *Ps* − *Pd*, *a*, *b* = coefficients [46,47]. Compared with *PWV*, CAVI is less affected by immediate blood pressure. To overcome the affection by immediate blood pressure, the stiffness parameter β, using an ultrasonic diagnostic apparatus, provides the proper arterial stiffness and is not affected by the blood pressure at measuring time.

#### 4.4.3. Measurement of Central Arterial Pressure (CAP)

CAP, which can be measured by invasive cardiac catheterization and noninvasive radial or carotid tension tests, but not brachial arterial blood pressure, predicts cardiovascular events in the elderly population. The augmentation index (AI) is obtained by detecting the pulse waveform of radial artery, which can provide the information of arterial elasticity, muscular arterial stiffness, and wave reflection. The increase of AI indicates arteriosclerosis. As heart rate can significantly affect the AI level, currently 75 beats/min heart rate is commonly used to correct the AI value. AI was a better predictor of cardiovascular events than systolic, diastolic, or pulse blood pressure. However, the common shortcomings of CAP and AI are the lack of reference values and limited research evidence, thus limiting the wide clinical application of these noninvasive vascular parameters.

### 4.5. Detection of Ankle Brachial Index (ABI)

There is an increasing number of patients with LEAD, which is caused by reduced blood flow in the blood vessels of the limb due to blockage or the narrowing of the blood vessels. The underlying causes of arterial disease are varied, with the most common being atherosclerosis. Other causes include vasculitis, vasospasm, embolism, thrombosis, myofibrodysplasia, or compartment syndrome. LEAD can range from atypical symptoms or intermittent claudication to severe limb ischemia (resting pain, ulcers, and gangrene), depending on the extent of obstruction of the blood vessels involved, and can lead to amputation without proper treatment. Measurement index: ABI. Calculation method: Divide the measured lower extremity systolic blood pressure from the upper extremity systolic blood pressure to obtain bilateral ABI. Reference range: normal low limit: ABI 0.91~0.99; Normal: ABI 1.00~1.30; Peripheral artery disease: ABI ≤ 0.9; Normal high limit: ABI 1.30~1.40; Arterial incompressibility or calcification: ABI > 1.4; Severity: Mild: ABI ≤ 0.90; Moderate: ABI ≤ 0.70; Severity: ABI ≤ 0.50. To increase the sensitivity of the approach, the European Society of Cardiology recommends measuring the postexercise ABI in patients with normal or low resting ABI and clinically suspected LEAD [48].

### 4.6. Analysis of Radial Artery Pulse Waveform

Measurement index: large arterial elasticity index C1 (mL/mmHg × 10), arteriolar elasticity index C2 (ml/mmHg × 100). Normal standard (for reference only): C1: 15~30 years old > 18 (ml/mmHg × 10); 31~45 years old > 16 (mL/mmHg × 10); 46~60 years old > 14 (mL/mmHg × 10); Over 60 years old > 10 (mL/mmHg × 10). C2: 15~30 years old > 8 (ml/mmHg × 100); 31~45 years old > 7 (mL/mmHg × 100); 46~60 years old > 6 (mL/mmHg × 100); Over 60 years old > 5 (mL/mmHg × 100).

The large artery elasticity index C1 and arteriolar elasticity index C2 obtained by detecting the pressure waveform of radial artery are also two parameters reflecting arterial elasticity. The study suggests that C1 and C2 have higher clinical value for the early identification of CVD than traditional blood pressure detection, and C1 and C2 are independently related to cardiovascular disease. However, whether the compliance obtained from the radial artery can reflect the compliance of the systemic blood vessels needs to be further confirmed. Similarly, the limited research evidence on C1 and C2 also limits the wide clinical application of these noninvasive vascular parameters [49].

### 4.7. Coronary Artery Calcification Score (CACS)

Vascular smooth muscle cell (VSMC) transformation to an osteochondrogenic phenotype is an initial step toward arterial calcification. Toll-like receptor 2 activates p38 and extracellular signal-regulated kinase 1/2 signaling to selectively modulate the upregulation of the IL-6-mediated receptor activator of nuclear factor κB ligand and the downregulation of osteoprotegerin. These signaling pathways act in concert to induce the chondrogenic transdifferentiation of VSMC, which in turn leads to vascular calcification during the pathogenesis of atherosclerosis [50]. Moreover, 5-MTP inhibits VSMC chondrocyte differentiation and calcification by blocking p38 MAPK-mediated CREB activation and IL-6 expression [51]. Osteopontin (OPN) was significantly and positively correlated to the coronary calcium score evaluated by cardiac computed tomography [52]. A cohort study that analyzed the effect of SPP1 polymorphisms in CVE in patients with chronic kidney disease found that the group of patients with cardiovascular events (CVE) had a higher incidence of atherosclerotic plaque and higher OPN levels at baseline. Single nucleotide polymorphisms (SNPs) in the rs1126616 SPP1 gene were independently associated with a higher incidence of CVE [53]. As a noninvasive method, computed tomography (CT) can detect the location and extent of coronary artery calcification. The long-term absence of calcification in coronary arteries is a healthy process of arterial aging. CACS can be used for the early diagnosis of coronary artery disease, treatment follow-up, CVD, and high-risk population screening. At present, the clinical evaluation of coronary artery calcification uses the Agatston integral system. When the CT value ≥ 130 HU (Hounsfield unit) and the area ≥ 1 mm^2^, calcification is considered to exist; that is, the Agatston integral = the pixel area of CT value ≥ 130 HU × CT peak coefficient of calcification foci. Among them, the CT value of calcification is 130~199 HU: CT peak coefficient of calcification is 1; Calcification CT value 200~299 HU: peak CT coefficient of calcification foci = 2; Calcification CT value 300~399 HU: peak CT coefficient of calcification foci = 3; Calcification CT value > 400 HU: peak CT coefficient of calcification foci = 4 [49]. The difference in endothelial function was apparent at high levels of CACS (>400) [54]. Its limitation is that there is no significant difference in endothelial function indicators in people with low calcification score levels.

### 4.8. Twenty-Four-Hour Ambulatory Blood Pressure and Ambulatory Electrocardiogram Monitoring

Twenty-four-hour ambulatory blood pressure monitoring can automatically and continuously record blood pressure changes within 24 h, including daytime and nighttime systolic blood pressure, diastolic blood pressure, mean blood pressure, maximum and minimum blood pressure, etc. Blood pressure variability directly reflects the stiffness and buffering function of blood vessels and is closely related to the occurrence of CVD, such as cognitive impairment [55,56]. Dynamic electrocardiogram (ECG) was first developed by Holter in 1957. It was originally used to monitor cardiac electrical activity, so it is also called the Holter electrocardiogram, which can continuously record ECG changes for 24 h, including heart rate, cardiac rhythm analysis, ST-T segment analysis, coronary artery ischemia status, and heart rate variability analysis to reflect the body’s vascular homeostasis. It is easy to operate, noninvasive, recommended by the guidelines to monitor its changes, and has been widely used in clinical practice.

### 4.9. Biomarker Detection of Vascular Aging and Endothelial Dysfunction

Soluble components of blood circulation, such as intercellular adhesion molecule-1, vascular cell adhesion molecule-1, and von Willebrand factor, can be used as markers of vascular senescence and endothelial dysfunction. Concentrations of these substances increase when endothelial cells are activated or damaged and predict the risk and severity of vascular disease. Other biomarkers of endothelial function also include endothelial progenitor cells [57], asymmetric dimethyl arginine, endothelial cells, particles, Rho kinase activity, nitrate/nitrite, E-select element, high-sensitivity C-reactive protein [58], interleukin-6 [58], endothelin-1 [59], serum albumin, blood clot regulatory proteins, fibrinolytic enzyme activators inhibitor-1, adiponectin, homocysteine [60], late-stage glycosylation end products, soluble advanced glycosylation end-product receptors, 8-hydroxydeoxyguanosine, F2-isoprostane, oxidized low-density lipoprotein, and microalbuminuria, etc. Biomarkers of vascular aging-related diseases also include blood lipid, blood glucose, uric acid, fibrinogen, triacylglycerol, B-type natriuretic peptide, D-dimer, glycated hemoglobin, and ceramide, which have been widely studied as risk factors of CVD. Different disease states have different vascular indicators and biomarkers and are affected by many factors. CAVI levels were higher in patients with hypertension, diabetes, CHD, and LEAD, and were associated with lipid levels, high-sensitivity C-reactive protein, B-type natriuretic peptide, and homocysteine levels. Studies in people without CVD have found that the use of high-sensitivity C-reactive protein or fibrin in the original assessment of risk can help prevent cardiovascular events. Therefore, the identification of new vascular disease-related biomarkers and timely lifestyle or drug intervention are of great significance to prevent or even reverse the occurrence of adverse cardiovascular events [61,62].

VWF is a glycoprotein involved in blood coagulation and mainly affects platelet adhesion. Serum vWF is mainly synthesized and secreted by endothelial cells. When endothelial cells are stimulated, the vWF stored inside the cells is released [63].

Endothelin-1, a peptide chain containing 21 amino acids, is used to contract blood vessels and increase blood pressure. Changes in shear forces, hypoxia, thrombin, and angiotensin II may stimulate additional endothelial secretion [64].

Thrombomodulin is a transmembrane glycoprotein of endothelial cell membrane. As a component of cell membrane, Thrombomodulin can bind thrombin to inhibit fibrin formation and platelet activation, and inhibit protein S inactivation, protein C activation, Va factor, and VIIIa factor inactivation. When vascular endothelial cells are damaged, thrombomodulin protease breaks down and is released into the blood, increasing TM levels by 1.5–2 times [65].

Osteopontin (OPN) is a matricellular protein that mediates diverse biological functions. Circulating OPN levels positively correlate with oxidative stress. A correlation linked OPN with the neutrophil degranulation biomarker MPO and resistin. OPN was identified as an independent variable associated with coronary artery disease in the low cardiovascular risk factor group [66]. Baseline high osteoprotegerin and OPN levels were independently associated with peripheral arterial disease (PAD) presence. Even higher levels of those biomarkers were detected among PAD patients with major adverse cardiovascular events [67].

Lipoprotein (a) [Lp(a)]-induced Notchl activation upregulated the BMP2-Smad1/5/9 pathway, resulting in cell calcification. Notchl activation also induced the translocation of nuclear factor-kappa B (NF-kappa B) accompanied by OPN overexpression and elevated inflammatory cytokines production. Lp(a) is a potential novel therapeutic target for vascular calcification [68].

Elevated blood homocysteine (Hcy) levels are strongly associated with endothelial dysfunction. Nanoscale selenium inhibits Hcy-induced mitochondrial oxidative damage and apoptosis by inhibiting the downregulation of glutathione peroxidase enzyme 1 and 4 (GPX1, GPX4) in vascular endothelial cells, and it can serve as a new strategy for the treatment of Hcy-mediated vascular dysfunction [60].

Human umbilical vein endothelial cells (HUVEC) can improve the cell survival, wound healing, migration, and angiogenesis of high-sugar-damaged HUVEC through paracrine signaling, with great potential in repairing diabetic vascular endothelial injury, but more research is needed [69].

The overexpression of miR-145 in an acute coronary syndrome rat model may improve the endothelial injury and abnormal inflammation through targeting FOXO1, suggesting it may be a therapeutic target [70]. Moreover, circ_0008360 knockdown reduced the high glucose-induced vascular endothelial dysfunction by regulating miR-186-5p and cyclin D2, suggesting that circ_0008360 might act as a target for the treatment of vascular endothelial dysfunction [71]. There should be uniform criteria for the method of identifying the gene profile of circRNAs before the peripheral blood circRNAs biomarker is translated into clinical practice. Large sample sizes are required to test their clinical effectiveness.

### 4.10. Genetic Evaluation

Arteriosclerosis is one of the markers of endothelial dysfunction, not the patent of modern people, as it existed earlier than modern civilization, and its risk factors and traditional risk factors are only part of the influencing factors of arteriosclerosis, and genetic factors also play an important part. Atherosclerosis is essentially age-based and begins after birth and progresses with age. Genes and environment play an important role in the occurrence and development of atherosclerosis, whereby genes create the vulnerability to atherosclerosis, and whether or when clinical atherosclerosis occurs is determined by environmental factors [49].

## 5. Establishment of China Vascular Health Comprehensive Evaluation System—BVHS Comprehensive Evaluation of Endothelial Function

In 2006, the first application guideline of vascular lesion detection technology in China was released. In 2011, the guideline was updated to add some new detection methods to comprehensively evaluate the structure and function of blood vessels from different aspects, such as biomarkers, imaging indicators, and noninvasive arteriosclerosis evaluation indicators. The comprehensive evaluation system of vascular senescence added the BVHS proposed by us in July 2015 [10]:

Grade I is normal: normal structure and function;

Grade II is arterial endothelial dysfunction: no radiographically proven atherosclerosis, FMD < 10%;

Grade III is arterial stiffness stage: no imaging confirmed atherosclerosis, CF-PWV > 9 m/s, CAVI > 9;

The early stage of grade IV is structural vascular disease: atherosclerotic lumen stenosis confirmed by imaging < 50%;

Grade IVa: atherosclerotic plaque formation, elastic normal (CF-PWV ≤ 9 m/s, CAVI ≤ 9);

Grade IVb: atheromatous plaque formation and decreased elasticity (CF-PWV > 9 m/s, CAVI > 9);

Grade V is structural vascular lesions;

Grade Va: atherosclerosis, lumen stenosis of 50%~75%, elastic normal (CF-PWV ≤ 9 m/s, CAVI ≤ 9);

Grade Vb: atherosclerosis, lumen stenosis of 50%~75%, reduced elasticity (CF-PWV > 9 m/s, CAVI > 9);

Advanced stage VI is structural vascular lesions: lumen stenosis > 75% (heart, brain, kidney, lower limb vessels);

Stage VII is a clinical vascular event stage (requiring emergency hospitalization): sudden vascular death, acute coronary syndrome, cerebrovascular accident, and lower limb artery occlusion.

BVHS integrates different evaluation and grading strategies, from vascular function to structure and from the early stage to the late stage of disease and carries out individualized intervention and management for people with different grades. Structural and functional indicators of vascular injury, which reflect the long-term cumulative effects of traditional and unidentified cardiovascular risk factors, can be considered as alternative endpoints for target organ injury before and even after clinical vascular events (Table 2).

The BVHS grading standard and its subsequent revision supplement the endothelial function was reflected by the RHI. Based on traditional risk factors, vascular structure and function indicators are combined, and blood vessels are directly used as evaluation targets, and the whole life cycle of the population is graded according to different vascular aging conditions [72]. BVHS has preliminarily verified that its value in predicting cardiovascular and cerebrovascular events is superior to traditional risk factor evaluation indexes [73]. The 2018 edition of the new Vascular health Assessment Guideline comprehensively introduces the existing vascular assessment indicators at home and abroad. Different vascular indicators, such as vascular endothelial function and arterial stiffness, can be obtained by different detection methods, but different indicators have different focuses which can be combined to complement each other. Different vascular indicators reflect different vascular health and aging status, so comprehensive evaluation is required [74].

BVHS focuses not only on vascular structural changes, such as stenosis or occlusion, but also on vascular function, such as elasticity and endothelial function, for comprehensive assessment. For people with vascular stenosis, subgroup management was further carried out. For those with vascular stenosis, the arterial elastic function was re-evaluated, and differentiated intervention strategies were adopted according to the elastic function status. Comprehensive evaluation of vascular senescence needs to integrate a variety of reliable vascular detection indicators. Given the focus on vascular structure and vascular function, the risk of cardiovascular and cerebrovascular events in patients with good vascular elasticity was lower. Various vascular structure and function indicators were used to manage vascular health by grades. Intervention strategies were different for different grades of people. No matter whether in clinical application or scientific research, attention should be paid to vascular aging as an evaluation target, and effective intervention measures should be taken before the occurrence of cardiovascular and cerebrovascular diseases. Promote vascular medicine and the management concept of the whole life cycle, and evaluate the health and aging of blood vessels as a whole. Vascular lesions at a certain site may be one of the signals of cardiovascular and cerebrovascular events, so attention should be paid. The application of the new BVHS standard provides different levels of population management and the practical implementation of the precision medicine model.

## 6. Implications of EndoFIND and Application Prospect of Vascular Endothelial Function Evaluation

Recently, we carried out the first large-scale multicenter clinical trial, the Endothelial Function Guided Therapy, in patients with nonobstructive coronary artery disease in the world (EndoFIND Study) (clinical trial registration time: 9 July 2019, registration number:NCT04013204). The rate, intensity, and long-term compliance of secondary preventive drug therapy, mainly based on the clinical practice of many cardiovascular patients, have always been unsatisfactory. Whether the introduction of noninvasive endothelial function testing into clinical diagnosis and treatment can increase doctors’ and patients’ attention to secondary prevention, thus improving the drug treatment rate, dose, and compliance of secondary prevention, and further reducing the risk of major adverse cardiovascular events, there is still no relevant clinical research evidence. In this study, patients with nonobstructive coronary artery disease were selected as subjects in the multicenter, randomized, blind, parallel controlled, two-stage clinical trial. The first stage was to evaluate whether the intervention significantly increased the prescribing rate of secondary prevention drugs. Phase II will further look at whether interventions can affect the risk of major adverse cardiovascular events based on the results of the phase I trial. Vascular or arterial disease is both the cause and the result. Therefore, effective, early and comprehensive assessment of vascular health status and the degree of vascular disease is of great significance for the occurrence of cardiovascular and cerebrovascular diseases in risk groups and the recurrence of acute adverse events in patients with cardiovascular and cerebrovascular diseases [75].

## 7. Implementation of Intelligent Vascular Health Management Strategy and Helping the “Healthy China 2030”

The Outline of the Healthy China 2030 Programme (the Platform) sets out the goal of achieving the effective control of major health risk factors by 2030. With vascular health management as the focus, people at risk of cardiovascular disease can be identified early and the risk of early intervention can be more conducive to the improvement of risk factors, thus helping to achieve the goal of controlling the main health risk factors.

The program aims to study and explore new technologies, methods, and strategies for the prevention and treatment of cardiovascular and cerebrovascular diseases in China based on vascular health management and vascular aging evaluation and information construction. Through the quality and efficient integration of existing health resources, it seeks to explore new types of medical and health service systems in combination with pension modes and implement the cardiovascular disease prevention and control of front moved forward, which will help to achieve “to the highest average life expectancy at 79.0 years old, 2030, a major chronic disease premature mortality 30% lower than in 2015”, the goal of the project.

Vascular senescence is the main cause of morbidity and mortality in the elderly population. It is necessary to understand the molecular and functional processes of vascular lesions to explore new effective measures to inhibit abnormal vascular aging. Pathophysiology associated with cellular and molecular mechanisms of the vascular system that accompanies the aging process, including oxidative stress, mitochondrial dysfunction, and molecular stressor response Decreased resistance, development of chronic low-level inflammation, genomic instability, cellular senescence, epigenetic alterations, loss of protein homeostasis, dystrophic induction disorders and stem cell dysfunction. The above-mentioned aging-related changes are the pathological basis of microvascular and macrovascular diseases [76,77]. It is of great significance to pay attention to the joint effect of aging and other cardiovascular risk factors on the vascular system, and to directly evaluate the physiological changes of vascular aging patients and the differences between the effects of other biological factors for the comprehensive cognition of vascular health. The early comprehensive maintenance of vascular health is also an important reflection of the moving forward of the “threshold” for the prevention of major chronic noncommunicable diseases in China. It will provide new ideas and new means for the early prevention and treatment of chronic diseases, and ultimately effectively prevent and control chronic diseases to achieve the national strategic goal of “healthy China”.

In China, the incidence of cardiovascular, brain, and LEAD is still increasing year-by-year, and even in recent years, the trend is becoming younger. Many people without obvious clinical symptoms, or young people who have not been diagnosed with cardiovascular and cerebrovascular diseases, also suffer from adverse cardiovascular events or even sudden death. Facing the severe situation of cardiovascular and cerebrovascular diseases, all health works in China have gradually shifted from focusing on disease treatment to disease prevention, disease rehabilitation, and life-cycle management. At present, there are many clinical studies related to blood vessels in foreign countries, while those in China are mostly basic, lacking large samples of high-quality clinical research data in the field of blood vessels, which needs to be further improved.

## 8. Cellular and Molecular Mechanism for Endothelial Function Protection

Diabetes mellitus is associated with endothelial dysfunction that leads to cardiovascular complications. SGLT2 inhibitors improve endothelial dysfunction by increasing NO and vasodilation, reducing inflammation, oxidative stress, and arterial stiffness, improving vascular remodeling, and inhibiting endothelial aging [78]. Distinct gene expression is associated with microvascular dysfunction [79].

## 9. Conclusions

Traditionally, a single measure of endothelial function has played a role in the management of cardiac and vascular health. Identifying new biomarkers related to cardiac and vascular diseases and applying a more scientific, comprehensive, and prospective cardiovascular health assessment system are of great significance for the early detection and intervention of diseases. The comprehensive assessment of all parameters of endothelial function, including FMD, RHI, PWV, CAVI, AI, genetic factors, and biomarkers, will contribute to full-cycle vascular health management.

## Figures and Tables

**Table 1 cells-11-03363-t001:** Pulse pressure and endothelial function and dysfunction.

Research Object	Result	Main Finding	References
Nonobstructed CAD	PP is inversely related to coronary microvascular function, as per noninvasive studies; invasive aortic PP is associated with coronary microvascular endothelial dysfunction and/or coronary microvascular endothelial independent dysfunction	High aortic PP was an independent predictor of all-cause mortality; coronary microvascular endothelial dysfunction was an independent risk factor in patients with low aortic PP	[29]
Age-related changes in vascular health and brain health	Primary aging and vascular risk factors are associated with increases in arterial stiffness and PP	Age-associated impairments in central arterial stiffness and peripheral vascular function have been attenuated or reversed through lifestyle behaviors such as exercise	[30]
Chronic kidney disease	The transmission of enhanced PP to the microcirculation leads to microvascular injury and organ dysfunction	Persistent coagulation activity is associated with aortic stiffness	[31]
Prediabetic individuals	The increase of brachial artery PP, central aortic PP, the decrease of reactive hyperemia index were significantly related to at least one PM pollutant	Glucose metabolic disorders may exacerbate vascular dysfunction associated with short-term ambient PM exposure	[32]
Patients with stable chronic heart failure	Increased β-OHB is associated with less empagliflozin-induced decrease in central PP	Increased β-OHB results in attenuated beneficial effects of empagliflozin on blood pressure and vascular parameters	[33]
Subjects undergoing hemodialysis.	Both PWV and PP decreased with higher D-Mg^2+^ compared to standard	Increasing dialysate magnesium improves vascular stiffness in subjects undergoing maintenance hemodialysis	[34]
Healthy volunteers	Phenylephrine increases PWV and PP more strongly than angiotensin II at the same mean aortic blood pressure	Acute changes in arterial stiffness has to bear in mind the superordinate integrative neural control of BP	[35]

**Table 2 cells-11-03363-t002:** Beijing Vascular Disease Evaluation Study.

Research Contents	Outcomes	References
Association of brain white matter lesions with arterial stiffness assessed by cardio-ankle vascular index	Higher arterial stiffness assessed by the cardio–ankle vascular index was associated with the presence of brain white matter lesions	[4]
Relationship between serum uric acid and vascular function and structure markers and gender difference in a real-world population of China	All vascular parameters were higher in males than females. There was no gender difference in the relationship between UA and vascular markers except in ABI. UA was independently linearly correlated with CAVI. In people with higher UA level, the risk of higher CF-PWV increased. Higher UA may influence the vascular function mainly instead of vascular structure.	[7]
The design and rationale of the Beijing Vascular Disease Patients Evaluation Study	A total of 2858 subjects were enrolled into our present study at baseline, and this present study will provide important information on the metabolic-related traditional and new risk factors, and establish a new vascular disease early detection system and scoring systems based on comprehensive vascular disease risk factors and vascular function and structure evaluation indexes	[8]
The relationship between serum bilirubin levels and peripheral arterial disease and gender difference in patients with hypertension	After adjusting for cardiovascular risk factors, PAD was independently negatively associated with TBiL and DBiL, and the association between PAD and bilirubin levels was only in men	[9]
Association between multisite atherosclerotic plaques and systemic arteriosclerosis: results from the BEST study	Higher systemic arteriosclerosis was independently associated with multisite atherosclerotic plaques (MAP), which indicate the supplementary value of arteriosclerosis for the earlier identification and intervention on MAP	[36]
Comparison of vascular-related diseases in their associations with carotid femoral pulse wave velocity	A study of a large clinical sample from Beijing, China confirmed a significant association of arteriosclerosis with different vascular-related diseases, including hypertension, diabetes, coronary heart disease, stroke, and PAD. Hypertension may be the single most important factor in arteriosclerosis	[41]
Relationship between glycated hemoglobin and low ankle–brachial index	HbA1c was an independent associated factor of lower ABI and linearly correlated to ABI level independent of fasting plasma glucose and other cardiovascular factors	[48]

## Data Availability

Not applicable.

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
