# Peer review of "Clinical Evaluation Tool for Vascular Health–Endothelial Function and Cardiovascular Disease Management"

_cells, 2022, doi:10.3390/cells11213363_

Round 1
Reviewer 1 Report
The manuscript by Wen et al., titled: "Clinical evaluation tool for vascular health-ENDOTHELIAL FUNCTION and cardiovascular disease management, reported a description of all recent technical advances for the clinical assessment for vascular endothelial function and vascular aging. The topic is very fascinating and interesting and the authors analyzed it in depth. The only concerns that should be addressed, is that there are several too long sentences throughout the manuscript leading to a reduction in its readibility. Thus, I suggest to revised them
Author Response
Point 1: The only concerns that should be addressed, is that there are several too long sentences throughout the manuscript leading to a reduction in its readibility.
Response 1: We have made correction according to the Reviewer’s comments. Several sentences in the manuscript that are too long have been revised.(such as 1, 2 and 4.1.1)
Special thanks to you for your good comments.
Reviewer 2 Report
Dear Authors,
It was a great pleasure to read your review on "Clinical evaluation tool for vascular health-ENDOTHELIAL FUNCTION and cardiovascular disease management". It is well-organized and written, relevant and original (as a review, of course). I have five minor comments: 1- Abstract would be more specific on which will be the review´s object; also some numerical data (on general health/diseases) in Pekin or China or in the world to increase the importance of the present manuscript; 2- A deeper analysis (not too long) should be added to Methods (on techniques) approaching the limitations of the vascular (and endothelial) function evaluation by dorsal hand vein technique (DHVT), sequential IMT (intima-media carotid, IMT), CAVI, CAP, FMD (flow-mediated dilation) and pulse wave velocity (PWV); 3- A few lines about inflammation and its markers ("biochemical markers") should be cited as well the limitations in clinical practice; 4- In the table 1. I think the 2K1C hypertension model is a rat/murine technique and I suggest the exclusion from this table; 5- All the references are from 2020-22-22 what is excellent but, in a review, some classical and historical papers should take a place.
Thanks for sharing this nice manuscript.
Reviewer 3 Report
This is well written manuscript
i would like to suggest.
1- 4.1.1 section when talking about PET it is interesting to expand on what tracers used as well as the role of MRI in more details especially in micrvascular dysfunction. You mention ECHO which im not sure the tte or toe can contribute. maybe focus on OCT and IVUS
2- 4.3 Can you expand on the role of PET in detecting inflaamtion as a marker of EC function considering the many tracers used in additon to traditional one such as 18F-NA and FDG.
3- 4.7 Calcification score. not sure what this tell us about EC function as this is in the whole body of vessels and CT is not suitable to look at EC. maybe you can link calcification to inflammation and role of MAPK/ p38/ OPN (SPP1) gene. There has been many recent publications with this regards
4- 4.9 Can you please expand and mention new potential biomarkers that been attracting interest such as OPN (SPP1) which is been tested as a biomarker of increased cardiovascular risk in asymptomatic patients.
Round 2
Reviewer 3 Report
No new comments to add